# Learning to Generate Reviews and Discovering Sentiment

## Abstract

We explore the properties of byte-level recurrent language models. When given sufficient amounts of capacity, training data, and compute time, the representations learned by these models include disentangled features corresponding to high-level concepts. Specifically, we find a single unit which performs sentiment analysis. These representations, learned in an unsupervised manner, achieve state of the art on the binary subset of the Stanford Sentiment Treebank. They are also very data efficient. When using only a handful of labeled examples, our approach matches the performance of strong baselines trained on full datasets. We also demonstrate the sentiment unit has a direct influence on the generative process of the model. Simply fixing its value to be positive or negative generates samples with the corresponding positive or negative sentiment.

## 1 Introduction and Motivating Work

Representation learning (Bengio et al. (2013)) plays a critical role in many modern machine learning systems. Representations map raw data to more useful forms and the choice of representation is an important component of any application. Broadly speaking, there are two areas of research emphasizing different details of how to learn useful representations.

The supervised training of high-capacity models on large labeled datasets is critical to the recent success of deep learning techniques for a wide range of applications such as image classification (Krizhevsky et al. (2012)), speech recognition (Hinton et al. (2012)), and machine translation (Wu et al. (2016)). Analysis of the task specific representations learned by these models reveals many fascinating properties (Zhou et al. (2014)). Image classifiers learn a broadly useful hierarchy of feature detectors re-representing raw pixels as edges, textures, and objects (Zeiler & Fergus (2014)). In the field of computer vision, it is now commonplace to reuse these representations on a broad suite of related tasks - one of the most successful examples of transfer learning to date (Oquab et al. (2014)).

There is also a long history of unsupervised representation learning (Olshausen & Field (1997)). Much of the early research into modern deep learning was developed and validated via this approach (Hinton & Salakhutdinov (2006); Huang et al. (2007); Vincent et al. (2008); Coates et al. (2010); Le (2013)). Unsupervised learning is promising due to its ability to scale beyond the small subsets and subdomains of data that can be cleaned and labeled given resource, privacy, or other constraints. This advantage is also its difficulty. While supervised approaches have clear objectives that can be directly optimized, unsupervised approaches rely on proxy tasks such as reconstruction, density estimation, or generation, which do not directly encourage useful representations for specific tasks. As a result, much work has gone into designing objectives, priors, and architectures meant to encourage the learning of useful representations. We refer readers to Goodfellow et al. (2016) for a detailed review.

Despite these difficulties, there are notable applications of unsupervised learning. Pre-trained word vectors are a vital part of many modern NLP systems (Collobert et al. (2011)). These representations, learned by modeling word co-occurrences, increase the data efficiency and generalization capability of NLP systems (Pennington et al. (2014); Chen & Manning (2014)). Topic modelling can also discover factors within a corpus of text which align to human interpretable concepts such as "art" or "education" (Blei et al. (2003)).

How to learn representations of phrases, sentences, and documents is an open area of research. Inspired by the success of word vectors, Kiros et al. (2015) propose skip-thought vectors, a method of training a sentence encoder by predicting the preceding and following sentence. The representation learned by this objective performs competitively on a broad suite of evaluated tasks. More advanced training techniques such as layer normalization (Ba et al. (2016)) further improve results. However, skip-thought vectors are still outperformed by supervised models which directly optimize the desired performance metric on a specific dataset. This is the case for both text classification tasks, which measure whether a specific concept is well encoded in a representation, and more general semantic similarity tasks. This occurs even when the datasets are relatively small by modern standards, often consisting of only a few thousand labeled examples.

In contrast to learning a generic representation on one large dataset and then evaluating on other tasks/datasets, Dai & Le (2015) proposed using similar unsupervised objectives such as sequence autoencoding and language modeling to first pretrain a model on a dataset and then finetune it for a given task. This approach outperformed training the same model from random initialization and achieved state of the art on several text classification datasets. Combining word-level language modelling of a dataset with topic modelling and fitting a small neural network feature extractor on top has also achieved strong results on document level sentiment analysis (Dieng et al. (2016)).

Considering this, we hypothesize two effects may be combining to result in the weaker performance of purely unsupervised approaches. Skip-thought vectors were trained on a corpus of books. But some of the classification tasks they are evaluated on, such as sentiment analysis of reviews of consumer goods, do not have much overlap with the text of novels. We propose this distributional issue, combined with the limited capacity of current models, results in representational underfitting. Current generic distributed sentence representations may be very lossy - good at capturing the gist, but poor with the precise semantic or syntactic details which are critical for applications.

The experimental and evaluation protocols may be underestimating the quality of unsupervised representation learning for sentences and documents due to certain seemingly insignificant design decisions. Hill et al. (2016) also raises concern about current evaluation tasks in their recent work which provides a thorough survey of architectures and objectives for learning unsupervised sentence representations - including the above mentioned skip-thoughts.

In this work, we test whether this is the case. We focus in on the task of sentiment analysis and attempt to learn an unsupervised representation that accurately contains this concept. Mikolov et al. (2013) showed that word-level recurrent language modelling supports the learning of useful word vectors. We are interested in pushing this line of work to learn representations of not just words but arbitrary scales of text with no distinction between sub-word, word, phrase, sentence, or document-level structure. Recent work has shown that traditional NLP task such as Named Entity Recognition and Part-of-Speech tagging can be performed this way by processing text as a byte sequence (Gillick et al. (2015)). Byte level language modelling is a natural choice due to its simplicity and generality. We are also interested in evaluating this approach as it is not immediately clear whether such a low-level training objective supports the learning of high-level representations. We train on a very large corpus picked to have a similar distribution as our task of interest. We also benchmark on a wider range of tasks to quantify the sensitivity of the learned representation to various degrees of out-of-domain data and tasks.

## 2 DATASET

Much previous work on language modeling has evaluated on relatively small but competitive datasets such as Penn Treebank (Marcus et al. (1993)) and Hutter Prize Wikipedia (Hutter (2006)). As discussed in Jozefowicz et al. (2016) performance on these datasets is primarily dominated by regularization. Since we are interested in high-quality sentiment representations, we chose the Amazon product review dataset introduced in McAuley et al. (2015) as a training corpus. In de-duplicated form, this dataset contains over 82 million product reviews from May 1996 to July 2014 amounting to over 38 billion training bytes. Due to the size of the dataset, we first split it into 1000 shards containing equal numbers of reviews and set aside 1 shard for validation and 1 shard for test.

## 3  MODEL AND TRAINING DETAILS

Many potential recurrent architectures and hyperparameter settings were considered in preliminary experiments on the dataset. Given the size of the dataset, searching the wide space of possible configurations is quite costly. To help alleviate this, we evaluated the generative performance in terms of log-likelihood of smaller candidate models after a single pass through the dataset and selected the best performing architecture according to this metric for large scale experiments. The model chosen is a single layer multiplicative LSTM (Krause et al. (2016)) with 4096 units.

We observed multiplicative LSTMs to converge faster than normal LSTMs for the hyperparameter settings that were explored both in terms of data and wall-clock time. The model was trained for a single epoch on mini-batches of 128 subsequences of length 256 for a total of 1 million weight updates. States were initialized to zero at the beginning of each shard and persisted across updates to simulate full-backpropagation and allow for the forward propagation of information outside of a given subsequence. Adam (Kingma & Ba (2014)) was used to accelerate learning with an initial 5e-4 learning rate that was decayed linearly to zero over the course of training. Weight normalization (Salimans & Kingma (2016)) was applied to the LSTM parameters. Data-parallelism was used across 4 Pascal Titan X gpus to speed up training and increase effective memory size. Training took approximately one month. The model is compact, containing approximately as many parameters as there are reviews in the training dataset. It also has a high ratio of compute to total parameters due to operating at a byte level. The model reaches 1.12 bits per byte.

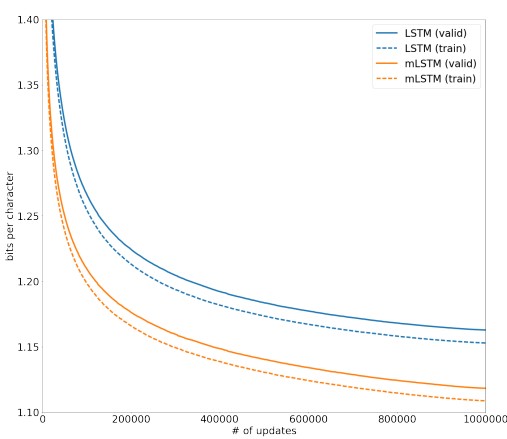

Figure 1: The mLSTM converges faster and achieves a better result within our time budget compared to a standard LSTM with the same hidden state size

## 4  EXPERIMENTAL SETUP AND RESULTS

Our model processes text as a sequence of UTF-8 encoded bytes (Yergeau (2003)). For each byte, the model updates its hidden state and predicts a probability distribution over the next possible byte. The hidden state of the model serves as an online summary of the sequence which encodes all information the model has learned to preserve that is relevant to predicting the future bytes of the sequence. We are interested in understanding the properties of the learned encoding. The process of extracting a feature representation is outlined as follows:

- Since newlines are used as review delimiters in the training dataset, all newline characters are replaced with spaces to avoid the model resetting state.

- Any leading whitespace is removed and replaced with a newline+space to simulate a start token. Any trailing whitespace is removed and replaced with a space to simulate an end token. HTML is unescaped and HTML tags are removed to mirror preprocessing applied to the training dataset. The text is encoded as a UTF-8 byte sequence.

- Model states are initialized to zeros. The model processes the sequence and the final cell states of the mLSTM are used as a feature representation.

We follow the methodology established in Kiros et al. (2015) by training a logistic regression classifier on top of our model's representation on datasets for tasks including semantic relatedness, text classification, and paraphrase detection. For the details on these comparison experiments, we refer the reader to their work. One exception is that we use an L1 penalty for text classification results instead of L2 as we found this performed better in the very low data regime.

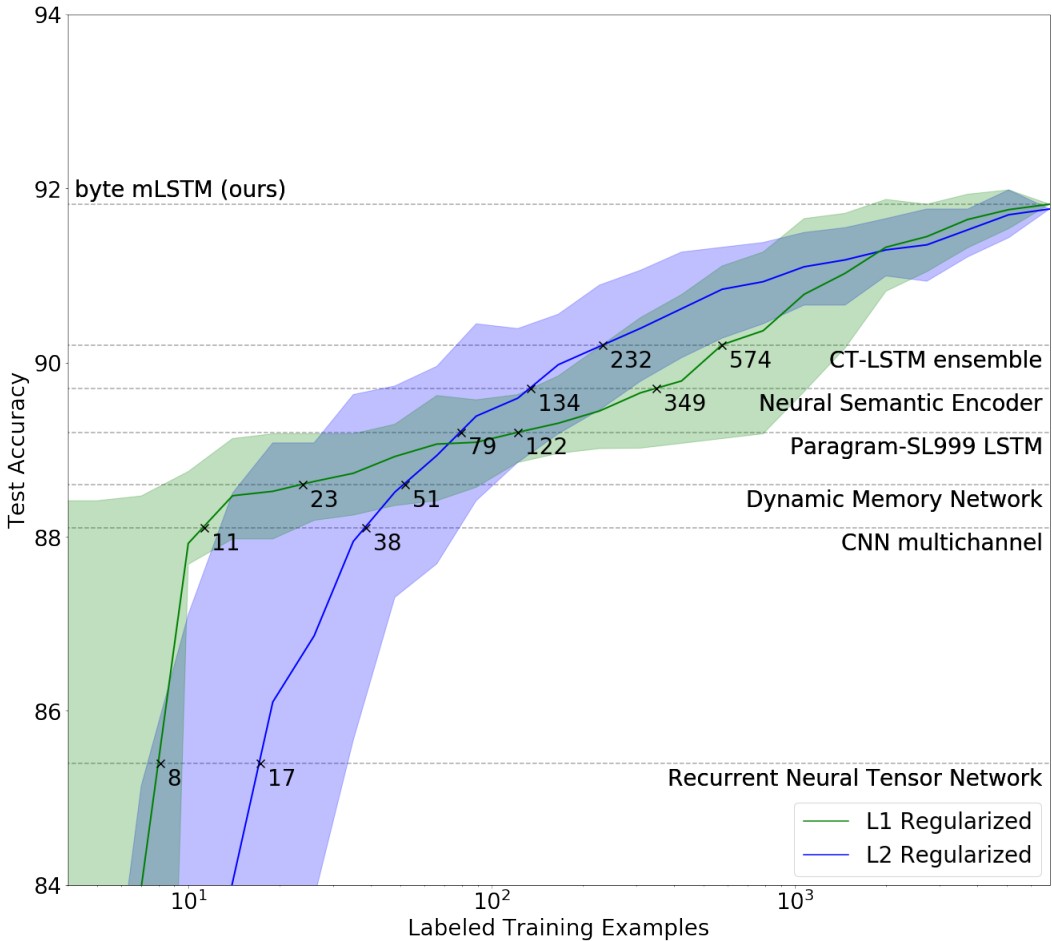

Figure 2: Performance on the binary version of SST as a function of labeled training examples. The solid lines indicate the average of 100 runs while the sharded regions indicate the 10th and 90th percentiles. Previous results on the dataset are plotted as dashed lines with the numbers indicating the amount of examples required for logistic regression on the byte mLSTM representation to match their performance. RNTN (Socher et al. (2013)) CNN (Kim (2014)) DMN (Kumar et al. (2015)) P-SL999 LSTM (Wieting et al. (2015)) NSE (Munkhdalai & Yu (2016)) CT-LSTM (Looks et al. (2017))

## 4.1 TEXT CLASSIFICATION AND SENTIMENT ANALYSIS

Table 1 shows the results of our model on 4 standard text classification datasets. The performance of our model is noticeably lopsided. On the MR (Pang & Lee (2005)) and CR (Hu & Liu (2004)) sentiment analysis datasets we improve the state of the art by a significant margin. The MR and CR datasets are sentences extracted from Rotten Tomatoes, a movie review website, and Amazon product reviews (which almost certainly overlaps with our training corpus). This suggests that our model has learned a rich representation of text from a similar domain. On the other two datasets, SUBJ's subjectivity/objectivity detection (Pang & Lee (2004)) and MPQA's opinion polarity (Wiebe et al. (2005)) our model has no noticeable advantage over other unsupervised representation learning approaches and is still outperformed by a supervised approach.

To better quantify the learned representation, we also test on a wider set of sentiment analysis datasets with different properties. The Stanford Sentiment Treebank (SST) (Socher et al. (2013)) was created specifically to evaluate more complex compositional models of language. It is derived from the same base dataset as MR but was relabeled via Amazon Mechanical Turk and includes dense labeling of the phrases of parse trees computed for all sentences. For the binary subtask, this amounts to

| Method | MR | CR | SUBJ | MPQA |
|---|---|---|---|---|
| NBSVM Wang & Manning (2012) | 79.4 | 81.8 | 93.2 | 86.3 |
| ST Kiros et al. (2015) | 77.3 | 81.8 | 92.6 | 87.9 |
| STLN Ba et al. (2016) | 79.5 | 83.1 | 93.7 | 89.3 |
| SDAE Hill et al. (2016) | 74.6 | 78.0 | 90.8 | 86.9 |
| CNN Kim (2014) | 81.5 | 85.0 | 93.4 | 89.6 |
| Adasent Zhao et al. (2015) | 83.1 | 86.3 | **95.5** | **93.3** |
| byte mLSTM | **86.9** | **91.4** | 94.6 | 88.5 |

Table 1: Small dataset classification accuracies

| Method | Err |
|---|---|
| FullBoW Maas et al. (2011) | 11.1 |
| NBSVM Mesnil et al. (2014) | 8.1 |
| **Sentiment unit (ours)** | 7.7 |
| SA-LSTM Dai & Le (2015) | 7.2 |
| **byte mLSTM (ours)** | 7.1 |
| TopicRNN Dieng et al. (2016) | 6.2 |
| Virt Adv Miyato et al. (2016) | 5.9 |

Table 2: IMDB sentiment analysis

76961 total labels compared to the 6920 sentence level labels. As a demonstration of the capability of unsupervised representation learning to simplify data collection and remove steps from a traditional NLP pipeline, our reported results ignore these dense labels and computed parse trees, using only the raw text and sentence level labels.

The representation learned by our model achieves 91.8% significantly outperforming the state of the art of 90.2% by a 30 model ensemble (Looks et al. (2017)). As visualized in Figure 2, our model is very data efficient. It matches the performance of baselines using as few as a dozen labeled examples and outperforms all previous results with only a few hundred labeled examples. This is under 10% of the total sentences in the dataset. Confusingly, despite a 16% relative error reduction on the binary subtask, it does not reach the state of the art of 53.6% on the fine-grained subtask, achieving 52.9%.

## 4.2 SENTIMENT UNIT

We conducted further analysis to understand what representations our model learned and how they achieve the observed data efficiency. The benefit of an L1 penalty in the low data regime (see Figure 2) is a clue. L1 regularization is known to reduce sample complexity when there are many irrelevant features (Ng (2004)). This is likely to be the case for our model since it is trained as a language model and not as a supervised feature extractor. By inspecting the relative contributions of features on various datasets, we discovered a single unit within the mLSTM *that directly corresponds to sentiment*. In Figure 3 we show the histogram of the final activations of this unit after processing IMDB reviews (Maas et al. (2011)) which shows a bimodal distribution with a clear separation between positive and negative reviews. In Figure 5 we visualize the activations of this unit on 6 randomly

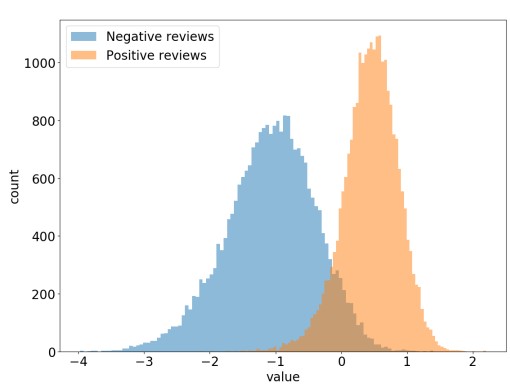

Figure 3: Histogram of cell values for the sentiment unit on IMDB reviews.

selected reviews from a set of 100 high contrast reviews which shows it acts as an online estimate of the local sentiment of the review. Fitting a threshold to this *single* unit achieves a test accuracy of 92.30% which outperforms a strong supervised results on the dataset, the 91.87% of NB-SVM trigram (Mesnil et al. (2014)), but is still below the semi-supervised state of the art of 94.09% (Miyato et al. (2016)). Using the full 4096 unit representation achieves 92.88%. This is an improvement of only 0.58% over the sentiment unit suggesting that almost all information the model retains that is relevant to sentiment analysis is represented in the very compact form of a single scalar. Table 2 has a full list of results on the IMDB dataset.

| Method | $r$ | $\rho$ | MSE |
|---|---|---|---|
| ST Kiros et al. (2015) | 0.858 | 0.792 | 0.269 |
| STLN Ba et al. (2016) | 0.858 | 0.788 | 0.270 |
| Tree-LSTM Tai et al. (2015) | **0.868** | **0.808** | **0.253** |
| byte mLSTM | 0.792 | 0.725 | 0.390 |

Table 3: SICK semantic relatedness subtask

| Method | Acc | F1 |
|---|---|---|
| ST Kiros et al. (2015) | 73.0 | 82.0 |
| SDAE Hill et al. (2016) | 76.4 | 83.4 |
| MTMETRICS Madnani et al. (2012) | **77.4** | **84.1** |
| byte mLSTM | 75.0 | 82.8 |

Table 4: Microsoft Paraphrase Corpus

### 4.3 CAPACITY CEILING

Encouraged by these results, we were curious how well the model's representation scales to larger datasets. We try our approach on the binary version of the Yelp Dataset Challenge in 2015 as introduced in Zhang et al. (2015). This dataset contains 598,000 examples which is an order of magnitude larger than any other datasets we tested on. When visualizing performance as a function of number of training examples in Figure 4, we observe a "capacity ceiling" where the test accuracy of our approach only improves by a little over 1% across a four order of magnitude increase in training data. Using the full dataset, we achieve 95.22% test accuracy. This better than a BoW TFIDF baseline at 93.66% but slightly worse than the 95.64% of a linear classifier on top of the 500,000 most frequent n-grams up to length 5.

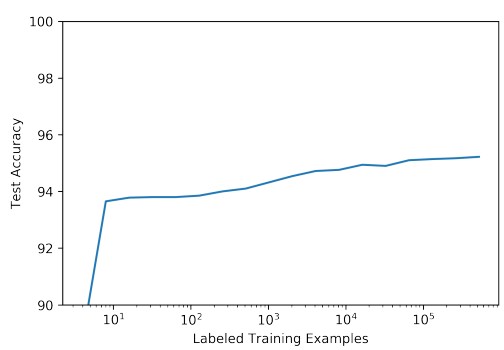

Figure 4: Performance on the binary version of the Yelp reviews dataset as a function of labeled training examples. The model's performance plateaus after about ten labeled examples and only slow improves with additional data.

The observed capacity ceiling is an interesting phenomena and stumbling point for scaling our unsupervised representations. We think a variety of factors are contributing to cause this. Since our model is trained only on Amazon reviews, it is does not appear to be sensitive to concepts specific to other domains. For instance, Yelp reviews are of businesses, where details like hospitality, location, and atmosphere are important. But these ideas are not present in reviews of products. Additionally, there is a notable drop in the relative performance of our approach transitioning from sentence to document datasets. This is likely due to our model working on the byte level which leads to it focusing on the content of the last few sentences instead of the whole document. Finally, as the amount of labeled data increases, the performance of a simple linear model trained on top of a static representation will eventually saturate. Complex models explicitly trained for a task can continue to improve and eventually outperform our approach when provided with enough labeled data.

With this context, the observed results reveal a natural ordering. On a small sentence level dataset from a very similar domain (the movie reviews of Stanford Sentiment Treebank) our model sets a new state of the art. But on a large, document level dataset from a slightly different domain (the Yelp reviews) it is only competitive with standard baselines.

### 4.4 REPRESENTATION STABILITY

The features of our model are learned via an unsupervised objective. A potential concern with this training methodology is that it is unreliable. There is no guarantee that the representations learned will be useful for a desired task because the model is not optimizing directly for this. This could lead to high variance of performance on desired tasks. This concern is further amplified for the properties of individual features and not just the representation as whole. While the work of Li et al. (2015) has shown that networks differing in only random initialization can learn the same features, this was

| Sentiment fixed to positive | Sentiment fixed to negative |
|---|---|
| Just what I was looking for. Nice fitted pants, exactly matched seam to color contrast with other pants I own. Highly recommended and also very happy! | The package received was blank and has no barcode. A waste of time and money. |
| This product does what it is supposed to. I always keep three of these in my kitchen just in case ever I need a replacement cord. | Great little item. Hard to put on the crib without some kind of embellishment. My guess is just like the screw kind of attachment I had. |
| Best hammock ever! Stays in place and holds it's shape. Comfy (I love the deep neon pictures on it), and looks so cute. | They didn't fit either. Straight high sticks at the end. On par with other buds I have. Lesson learned to avoid. |
| Dixie is getting her Doolittle newsletter we'll see another new one coming out next year. Great stuff. And, here's the contents - information that we hardly know about or forget. | great product but no seller. couldn't ascertain a cause. Broken product. I am a prolific consumer of this company all the time. |
| I love this weapons look . Like I said beautiful !!! I recommend it to all. Would suggest this to many roleplayers , And I stronge to get them for every one I know. A must watch for any man who love Chess! | Like the cover, Fits good. . However, an annoying rear piece like garbage should be out of this one. I bought this hoping it would help with a huge pull down my back & the black just doesn't stay. Scrap off everytime I use it.... Very disappointed. |

Table 5: Random samples from the model generated when the value of sentiment unit hidden state is fixed to either -1 or 1 for all steps. The sentiment unit has a strong influence on the model's generative process.

tested for CNNs trained for supervised image classification. To check whether this is the case for our approach, we trained a second model using the same hyperparameters differing only in weight parameterization and compared to the model discussed in this paper. The best single unit in this model (selected via validation set) achieves 92.42% test accuracy on IMDB compared to the 92.30% of the unit visualized in this paper. This suggests that the sentiment unit is an example of a convergent representation.

## 4.5 OTHER TASKS

Besides classification, we also evaluate on two other standard tasks: semantic relatedness and paraphrase detection. While our model performs competitively on Microsoft Research Paraphrase Corpus (Dolan et al. (2004)) in Table 3, it performs poorly on the SICK semantic relatedness task (Marelli et al. (2014)) in Table 4. It is likely that the form and content of the semantic relatedness task, which is built on top of descriptions of images and videos and contains sentences such as "A sea turtle is hunting for fish" is effectively out-of-domain for our model which has only been trained on the text of product reviews.

## 4.6 GENERATIVE ANALYSIS

Although the focus of our analysis has been on the properties of our model's representation, it is trained as a generative model and we are also interested in its generative capabilities. Hu et al. (2017) and Dong et al. (2017) both designed conditional generative models to disentangle the content of text from various attributes like sentiment or tense. We were curious whether a similar result could be achieved using the sentiment unit. In Table 5 we show that by simply setting the sentiment unit to be positive or negative, the model generates corresponding positive or negative reviews. While all sampled negative reviews contain sentences with negative sentiment, they sometimes contain sentences with positive sentiment as well. This might be reflective of the bias of the training corpus which contains over 5x as many five star reviews as one star reviews. Nevertheless, it is interesting to see that such a simple manipulation of the model's representation has a noticeable effect on its behavior. The samples are also high quality for a byte level language model and often include valid sentences.

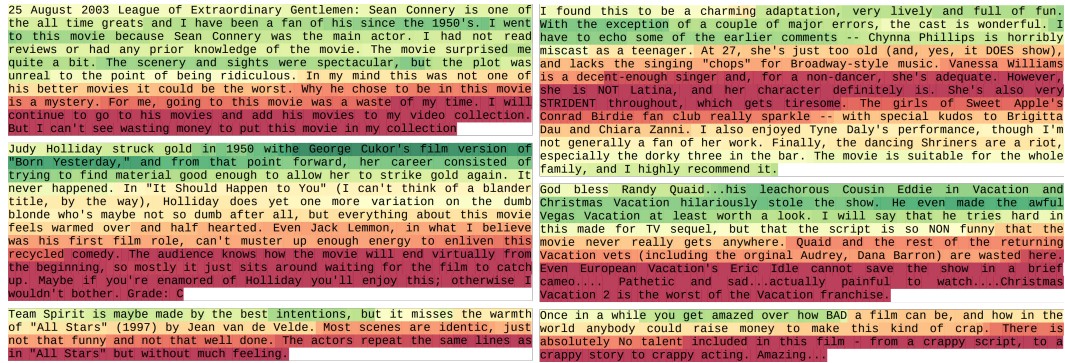

Figure 5: Visualizing the value of the sentiment unit's cell state as it processes six randomly selected high contrast IMDB reviews. Red indicates negative sentiment while green indicates positive sentiment. Best seen in color.

## 5 DISCUSSION AND FUTURE WORK

It is an open question why our model recovers the concept of sentiment in such a precise, disentangled, interpretable, and manipulable way. It is possible that sentiment as a conditioning feature has strong predictive capability for language modelling. This is likely since sentiment is such an important component of a review. Previous work analyzing LSTM language models showed the existence of interpretable units that indicate position within a line or presence inside a quotation (Karpathy et al. (2015)). In many ways, the sentiment unit in this model is just a scaled up example of the same phenomena. The update equation of an LSTM could play a role. The element-wise operation of its gates may encourage axis-aligned representations. Models such as word2vec have also been observed to have small subsets of dimensions strongly associated with specific tasks (Li et al. (2016)).

Our work highlights the sensitivity of learned representations to the data distribution they are trained on. The results make clear that it is unrealistic to expect a model trained on a corpus of books, where the two most common genres are Romance and Fantasy, to learn an encoding which preserves the exact sentiment of a review. Likewise, it is unrealistic to expect a model trained on Amazon product reviews to represent the precise semantic content of a caption of an image or a video.

There are several promising directions for future work highlighted by our results. The observed performance plateau, even on relatively similar domains, suggests improving the representation model both in terms of architecture and size. Since our model operates at the byte-level, hierarchical/multi-timescale extensions could improve the quality of representations for longer documents. The sensitivity of learned representations to their training domain could be addressed by training on a wider mix of datasets with better coverage of target tasks. Finally, our work encourages further research into language modelling as it demonstrates that the standard language modelling objective with no modifications is sufficient to learn high-quality representations.

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
