# OpenReview forum: "Learning To Generate Reviews and Discovering Sentiment"
_ICLR.cc/2018/Conference — Reject_

### Official Review · AnonReviewer3 · 2017-11-27
**Interesting experiments but lack of model description**

**Rating:** 4
**Confidence:** 3

**Review:**

The authors propose to use a byte level RNN to classify reviews. In the meantime, they learn to generate reviews. The authors rely on the multiplicative LSTM proposed by Krause et al. 2016, a generative model predicting the next byte. They apply this architecture on the same task as the original article: document classification; they use a logistic regression on the extracted representation. The authors propose an evaluation on classical datasets and compare themselves to the state of the art.
The authors obtain interesting results on several datasets. They also explore the core of the unsupervised architecture and discover a neuron which activation matches the sentiment target very accurately. A deeper analyze shows that this neuron is more efficient on small datasets than on larger.
Exploiting the generative capacity of the network, they play with the "sentiment neuron" to deform a review. Qualitative results are interesting.




The authors do not propose an original model and they do not describe the used model inside this publication.

Nor the model neither the optimized criterion is detailled: the authors present some curve mentioning "bits per character" but we do not know what is measured. In fact, we do not know what is given as input and what is expected at the output -some clues are given in the experimental setup, but not in the model description-.

Figure 2 is very interesting: it is a very relevant way to compare authors model with the literature.

Unfortunately, the unsupervised abilities of the network are not really explained: we are a little bit frustrated by section 5.

==

This article is very interesting and well documented. However, according to me, the fact that it provides no model description, no model analysis, no modification of the model to improve the sentiment discovery, prevents this article from being publicized at ICLR.

---

### Official Review · AnonReviewer2 · 2017-11-28
**This paper needs more serious work**

**Rating:** 2
**Confidence:** 5

**Review:**

First of all, I don't think I fully understand this paper, because it is difficult for me to find answers from this paper to the following questions:
1) what is the hypothesis in this paper? Section 1 talks about lots of things, which I don't think is relevant to the central topic of this paper. But it misses the most important thing: what is THIS paper (not some other deep learning/representation problems)
2) about section 2, regardless whether this is right place to talk about datasets, I don't understand why these two datasets. Since this paper is about generating reviews and discovering sentiment (as indicated in the paper)
3) I got completely confused about the content in section 3 and lost my courage to read the following sections.

---

### Official Review · AnonReviewer1 · 2017-11-28

**Rating:** 4
**Confidence:** 5

**Review:**

This paper shows that an LSTM language model trained on a large corpus of Amazon product reviews can learn representations that are useful for sentiment analysis.
Given representations from the language model, a logistic regression classifier is trained with supervised data from the task of interest to produce the final model.
The authors evaluated their approach on six sentiment analysis datasets (MR, CR, SUBJ, MPQA, SST, and IMDB), and found that the proposed method is competitive with existing supervised methods.
The results are mixed, and they understandably are better for test datasets from similar domains to the Amazon product reviews dataset used to train the language model.
An interesting finding is that one of the neurons captures sentiment property and can be used to predict sentiment as a single unit.

I think the main result of the paper is not surprising and does not show much beyond we can do pretraining on unlabeled datasets from a similar domain to the domain of interest.
This semi-supervised approach has been known to improve in the low data regime, and pretraining an expressive neural network model with a lot of unlabeled data has also been shown to help in the past.
There are a few unanswered questions in the paper:
- What are the performance of the sentiment unit on other datasets (e.g., SST, MR, CR)? Is it also competitive with the full model?
- How does this method compare to an approach that first pretrains a language model on the training set of each corpus without using the labels, and then trains a logistic regression while fixing the language model? Is the large amount of unlabeled data important to obtain good performance here? Or is similarity to the corpus of interest more important?
- I assume that the reason to use byte LSTM is because it is cheaper than a word level LSTM. Is this correct or was there any performance issue with using the word directly?
- More analysis on why the proposed method does well on the binary classification task of SST, but performs poorly on the fine-grained classification would be useful. If the model is capturing sentiment as is claimed by the authors, why does it only capture binary sentiment instead of a spectrum of sentiment level?

The paper is also poorly written. There are many typos (e.g., "This advantage is also its difficulty", "Much previous work on language modeling has evaluated ", "We focus in on the task", and others) so the writing needs to be significantly improved for it to be a conference paper, preferably with some help from a native English speaker.

---

### Public Comment · ~Cuong_Hoang1 · 2017-11-08
**Cool work, there are several very interesting findings but the paper could be improved**

I am not an expert by any means. So I simply put my comments on the paper here, and hope someone would correct me if I was wrong at some points.

I found this paper interesting because of two things. First, several things from the paper are new to me. Second, I like the way the authors make a very good story from their experiments.

Unsupervised representation learning is very promising since unlabeled data are every where. But to date, supervised learning models still outperform unsupervised models. This may be explained because "supervised approaches have clear objectives that can be directly optimized". Meanwhile, "unsupervised approaches rely on proxy tasks such as reconstruction, density estimation, or generation, which do not directly encourage useful representations for specific tasks." The paper exploits other perspectives: distributional issue and the limited capacity of current unsupervised representation learning models. Specifically, "current generic distributed sentence representations may be very lossy - good at capturing the gist, but poor with the precise semantic or syntactic details which are critical for applications." This combines with the limited capacity may be the root of devil, and the authors investigate into details this point.

How? The authors first attempts to learn an unsupervised representation by training byte (character) level language modelling. Then we can use the outputs to train a sentiment analysis classifier. The authors trained their model on a very large dataset (Amazon review dataset) (the training took 1 month!)

Given a new text (paragraph, article or whatever), we simply perform some pre-processing and then feed the text into the mLSTM. Here is the interesting thing: we then get the outputs of all the output units (there are 4,096 units) and consider them as a feature vector representing the string read by the model. We turned the model into a sentiment classifier by taking a linear combination of these units, learning the weights of the combination via the available supervised data. This is new to me, indeed.

What is next? By inspecting the relative contributions of features, they discovered a single unit within the LSTM that directly corresponds to sentiment. This is a very surprising finding, as remember that the mLSTM model is trained only to predict the next character in text.

But why is it the case? It is indeed an open question why the model recovers the concept of sentiment in such a precise way. It is pity, however, that the authors don't dig into details to have a satisfied answer!

Overall I like the paper and like their interesting findings. This is a very cool work!

But I think the paper could be significantly improved in two ways:

- I don't think the story written in the paper is really coherent.

- The findings are interesting but a deeper investigation would satisfy readers more. So far everything is still as "I read a very cool paper which shows that there exists a neural sentiment neuron by simply training language modeling, but I don't know why!".

---

### Decision · Program_Chairs · 2018-01-29
**ICLR 2018 Conference Acceptance Decision**

**Decision:**

Reject

**Comment:**

The paper reports experiments where a LSTM language model is pretrained on a large corpus of reviews, and then the produced representation is used within a classifier on a number of sentiment classification datasets.  The relative success of the method is not surprising. The novelty is very questionable, the writing quality is mixed (e.g., typos, the model is not even properly described). There are many gaps in evaluation (e.g., from the intro it seems that the main focus is showing that byte level modeling is preferable to more standard set-ups -- characters / BPE / words). However, there are (almost) no experiments supporting this claim. The same is true for the 'sentiment neuron': its effectiveness is also not properly demonstrated. In general, the results are somewhat mixed.

Pros:
-- good results on some datasets
Cons:
-- limited novelty
-- some claims are not tested / issues with evaluation
-- writing quality is not sufficient / clarity issues


Overall, the reviewers are in agreement that the paper does not meet ICLR standards.